# Evidence for Pentapeptide-Dependent and Independent CheB Methylesterases

**DOI:** 10.3390/ijms21228459

**Published:** 2020-11-11

**Authors:** Félix Velando, José A. Gavira, Miriam Rico-Jiménez, Miguel A. Matilla, Tino Krell

**Affiliations:** 1Department of Environmental Protection, Estación Experimental del Zaidín, Consejo Superior de Investigaciones Científicas, Prof. Albareda 1, 18008 Granada, Spain; felix.velando@eez.csic.es (F.V.); miriamrj@gmail.com (M.R.-J.); 2Laboratory of Crystallographic Studies, IACT, (CSIC-UGR), Avenida de las Palmeras 4, 18100 Armilla, Spain; jgavira@iact.ugr-csic.es

**Keywords:** bacterial signal transduction, chemosensory pathways, chemoreceptor, X-ray structure, C-terminal pentapeptide, CheB, methylesterase

## Abstract

Many bacteria possess multiple chemosensory pathways that are composed of homologous signaling proteins. These pathways appear to be functionally insulated from each other, but little information is available on the corresponding molecular basis. We report here a novel mechanism that contributes to pathway insulation. We show that, of the four CheB paralogs of *Pseudomonas aeruginosa* PAO1, only CheB_2_ recognizes a pentapeptide at the C-terminal extension of the McpB (Aer2) chemoreceptor (*K*_D_ = 93 µM). McpB is the sole chemoreceptor that stimulates the Che2 pathway, and CheB_2_ is the methylesterase of this pathway. *Pectobacterium atrosepticum* SCRI1043 has a single CheB, CheB_Pec, and 19 of its 36 chemoreceptors contain a C-terminal pentapeptide. The deletion of *cheB_Pec* abolished chemotaxis, but, surprisingly, none of the pentapeptides bound to CheB_Pec. To determine the corresponding structural basis, we solved the 3D structure of CheB_Pec. Its structure aligned well with that of the pentapeptide-dependent enzyme from *Salmonella enterica*. However, no electron density was observed in the CheB_Pec region corresponding to the pentapeptide-binding site in the *Escherichia coli* CheB. We hypothesize that this structural disorder is associated with the failure to bind pentapeptides. Combined data show that CheB methylesterases can be divided into pentapeptide-dependent and independent enzymes.

## 1. Introduction

Chemosensory pathways are among the most abundant prokaryotic signal transduction mechanisms [1]. Apart from mediating flagellum based chemotaxis, chemosensory pathways carry out alternative cellular functions like the control of second messenger levels or type IV pili-based motility [1,2,3]. The key element of a chemosensory pathway is the ternary complex formed by chemoreceptors, the CheA autokinase and the CheW coupling protein. Signaling is typically initiated by signal recognition at the chemoreceptor ligand-binding domain (LBD) that creates a molecular stimulus modulating CheA autophosphorylation and, subsequently, transphosphorylation to the CheY response regulator. The ratio of CheY to phosphorylated CheY (CheY-P) defines the pathway output [4].

The pathway sensitivity is adjusted by the coordinated action of the CheR methyltransferase and the CheB methylesterase that catalyze methylation and demethylation, respectively, of several glutamate residues at the chemoreceptor-signaling domain. It was shown that poorly methylated chemoreceptors have high chemoeffector affinity and high propensity for methylation [4,5]. Genome analyses revealed that genes encoding both enzymes are present in the large majority of chemosensory pathways and are thus among the six core pathway proteins [1].

In *Escherichia coli*, CheR was found to bind to the methylation site of the Tar chemoreceptor with a rather modest affinity, ranging between 100 to 200 µM, depending on experimental conditions [6]. However, Tar possesses a C-terminal pentapeptide that is tethered to the C-terminal end of the chemoreceptor signaling domain via an unstructured linker [7]. This pentapeptide, NWETF, represents an additional binding site for CheR and CheB [8,9]. CheR from *E. coli* bound NWETF with a *K*_D_ value affinity of approximately 2 µM, an affinity that is around 100-fold higher than the affinity for the methylation site [6,10]. It was proposed that CheR binding to the pentapeptide enhanced the local CheR concentration, leading to the optimal adaptation [11,12]. Remarkably, genome analyses indicated that approximately 10% of chemoreceptors possess a C-terminal pentapeptide [9], and experimental studies, as well as sequence analyses of CheR from different species, have shown that this protein family can be subdivided into pentapeptide-dependent and independent enzymes that are either able or unable to bind these pentapeptides [8,13]. The structure of the CheR-pentapeptide complex has been solved [14], and several sequence features at or close to the pentapeptide-binding site have been identified to be specific for each CheR subfamily [8,13].

Compared to CheR, less information is available on CheB function. Much of what we know of this protein family is due to the studies of the enzymes from *E. coli* [15,16,17,18,19] and *Salmonella enterica* sv. Typhimurium [20,21,22,23] that share 95% of their amino acid sequence identity (Appendix A). Both species possess a single chemosensory pathway, a single pentapeptide-dependent CheB and two chemoreceptors, Tar and Tsr, that contain a C-terminal pentapeptide. CheB from *E. coli* bound this pentapeptide with much lower affinity (*K*_D_ = 130 to 160 µM, depending on the method used) [17] as compared to CheR. The affinity of CheB for the pentapeptide is thus too low to increase the local concentration, but the CheB-pentapeptide interaction was found to stimulate methylesterase activity [17]. CheB is composed of a phosphoryl group-accepting receiver domain (REC) and a methylesterase domain [20], and it was shown that REC domain phosphorylation stimulates the catalytic activity of CheB [17].

The mutation or removal of the NWETF pentapeptide from the *E. coli* Tar and Tsr chemoreceptors largely reduced methylation and demethylation *in vivo* and *in vitro* and abolished chemotaxis [11,15,19,24,25]. This results in the paradoxical situation where C-terminal pentapeptides are essential for the functioning of some receptors, like Tar and Tsr, but are absent from many other chemoreceptors that mediate strong chemotactic responses [26,27,28,29,30]. Consequently, the physiological relevance of pentapeptide-dependent chemosignaling remains to be established, but we recently showed that pentapeptide-containing chemoreceptors are more abundant in bacteria that maintain host interactions [9]. In addition, it is only a little clear to which degree there are pentapeptide-independent CheB. To address this latter issue, we report here studies of CheB homologs from the human and plant pathogenic bacterial strains *Pseudomonas aeruginosa* PAO1 [31] and *Pectobacterium atrosepticum* SCRI1043 [32].

*P. aeruginosa* PAO1 has five gene clusters encoding signaling proteins that assemble into four chemosensory pathways (Figure 1) [33].

These pathways differ in function: whereas the Che pathway mediates chemotaxis [34,35], the Wsp pathway controls the c-di-GMP levels [2], and the Chp pathway was associated with type IV pili-mediated motility and cAMP levels [3,36]. The function of the Che_2_ pathway is unknown. As shown in Figure 1, each chemosensory pathway contains a CheR and CheB homolog. Experimental and bioinformatic studies indicate that, of the 26 chemoreceptors, McpB (synonym Aer2) (Figure 1) is the sole chemoreceptor that feeds into the Che_2_ pathway [33,37]. Furthermore, it is the only chemoreceptor with a C-terminal pentapeptide [13]. We previously showed that, of the four CheR homologs, the methyltransferase of the Che_2_ pathway, CheR_2_, was the only homolog that bound the McpB pentapeptide [13]. Furthermore, binding to this pentapeptide was essential for the CheR_2_ interaction and methylation of McpB [13]. We concluded that the specific pentapeptide-CheR_2_ interaction is a mechanism that permits the targeting a particular chemoreceptor with a specific CheR [13].

The four CheB homologs of *P. aeruginosa* were found to play important physiological roles. A *cheB*_1_ mutant was nonchemotactic [38], and screening of a 2200-member mutant library for virulence defects in a cystic fibrosis airway *P. aeruginosa* isolate revealed that a *cheB*_2_ mutant showed one of the strongest phenotypes in *Caenorhabditis elegans*, a finding that was confirmed by experimentation on mice [39]. A *cheB*_3_/*wspF* mutant caused elevated c-di-GMP levels and enhanced biofilm formation [2], due to locking this pathway into an active state. Interestingly, cystic fibrosis airway infections frequently produce rugose small-colony variants (RSCV), and this phenotype could be reverted by the *in trans* expression of *cheB*_3_/*wspF*, indicating that *cheB*_3_/*wspF* mutations are a very frequent mechanism for generating RSCV morphotypes [40]. As for the *cheB*_4_ gene, its mutation caused hyper-piliation [3], prevented swarming, and formed more robust biofilms by stimulating matrix production [41]. Considering the pentapeptide-mediated specific interaction of CheR_2_ and McpB, we studied here the interaction of the McpB pentapeptide with the four CheB homologs.

Since *P. aeruginosa* has four CheB paralogs and a single pentapeptide-containing chemoreceptor, we aimed at studying the inverse situation, i.e., a bacterium with a single CheB but multiple chemoreceptors with pentapeptides of different sequences. To this end, we chose *P. atrosepticum* SCRI1043 as a model that has a single CheB and 19 chemoreceptors with a pentapeptide [32]. *P. atrosepticum* is among the top 10 plant pathogens [42] and the causative agent of soft rot diseases in many agriculturally relevant crops [43]. This species belongs, like *E. coli* and *S. enterica*, to the *Enterobacteriacae* family. Combined data from both strains allows distinguishing between pentapeptide-dependent and independent CheB proteins.

## 2. Results

### 2.1. P. aeruginosa CheB_2_ Is the Only CheB Homolog That Binds to the McpB Chemoreceptor Pentapeptide

We reported previously that CheR_2_ is the only one of the four *P. aeruginosa* CheR homologs that binds the terminal pentapeptide of the McpB chemoreceptor [13]. To verify which CheB homolog interacts with this peptide, the four CheB homologs were overexpressed in *E. coli*, purified, and submitted to microcalorimetric titrations with the pentapeptide of the McpB receptor, GWEEF. The titration of CheB_1_, CheB_3_, and CheB_4_ with the peptide caused small and uniform heat changes that were similar to ligand dilution heats (Figure 2). In contrast, exothermic binding heats were observed for the titration of CheB_2_, and a dissociation constant (*K*_D_) of 93 ± 15 µM was derived.

This indicated that the CheR and CheB homologs of the Che_2_ pathway specifically interact with the only chemoreceptor that feeds into this pathway, McpB (Figure 1). The measured affinity was approximately 180-fold lower than that for the peptide binding to CheR_2_ [13] (*K*_D_ = 0.52 µM), implying that CheR_2_ largely outcompetes CheB_2_ for binding at the GWEEF pentapeptide. Next, it was investigated whether CheB_2_ phosphorylation alters the affinity for the pentapeptide. Since the phosphorylation half-life of CheB proteins is typically very short [44], we generated stable beryllium fluoride adducts that mimic phosphorylation [45]; however, protein precipitation made any biochemical study impossible. Previous studies showed that the replacement of the phosphoryl group accepting aspartate with glutamate in receiver domains mimics protein phosphorylation [46]. We generated the CheB_2_ D55E mutant protein that was titrated with the GWEEF peptide, resulting in a *K*_D_ of 56 ± 14 µM (Appendix A), representing a modest increase in affinity as compared to the native protein.

### 2.2. The Signaling Gene Cluster of P. atrosepticum SCRI1043 Encodes a Chemosensory Pathway That Mediates Chemotaxis

Chemosensory pathways can exert a number of different functions like chemotaxis, modulating second messenger levels and type IV pili-based movement [1]. To explore the function of the sole chemosensory gene cluster in *P. atrosepticum* SCRI1043 (Figure 1), we created deletion mutants of the *cheA* and *cheB* genes. Subsequently, we conducted quantitative capillary chemotaxis assays of the wild-type (wt) and mutant strains towards casamino acids. As shown in Figure 3, the wt strain showed strong chemotactic responses, whereas the *cheA* and *cheB* mutants failed to respond, indicating that the chemosensory pathway mediates chemotaxis.

### 2.3. P. atrosepticum SCRI1043 Contains a Large Number of Chemoreceptors with a C-Terminal Pentapeptide

The chemoreceptor repertoire of *P. atrosepticum* SCRI1043 is illustrated in Figure 4. Previous studies have shown that approximately 14% of bacterial chemoreceptors lack transmembrane regions and are thus involved in the sensing of cytoplasmic signals [47]. No such receptors are present in *P. atrosepticum* SCRI1043, since all 36 chemoreceptors are membrane-bound and possess two transmembrane regions (Figure 4).

There are three receptors that possess the typical topology and domain arrangement of Aer receptors [49] that mediate aerotaxis (Figure 4). Only two receptors possess a dCache type LBD that are highly abundant sensor domains in chemoreceptors and sensor kinases [50] and respond mainly to different amines [26]. In addition, the repertoire contains one and four receptors with sCache or HBM LBDs, respectively, that are typically organic acid sensors [51]. The alignment of the C-terminal segment of *P. atrosepticum* SCRI1043 chemoreceptors revealed that 19 of them possess a terminal pentapeptide that is tethered to the signaling domain via linker sequences of 29 to 39 amino acids (Figure 5).

In total, there were nine different pentapeptide sequences—among which, NWETF, the pentapeptide of the *E. coli* and *S. enterica* sv. Typhimurium receptors, was the most abundant and present in eight chemoreceptors (Appendix A). Interestingly, the 19 pentapeptide-containing chemoreceptors possess either 4HB or HBM-type LBDs that correspond to either single- or double-module four-helical bundle domains, respectively (Figure 4). The linkers showed no apparent sequence similarities (Appendix A) and were predicted to be mainly unstructured (Appendix A).

### 2.4. P. atrosepticum SCRI1043 CheB Fails to Recognize Pentapeptides

We subsequently overexpressed and purified *P. atrosepticum* CheB (CheB_Pec) to study its interaction with the pentapeptides present in SCRI1043 chemoreceptors. However, microcalorimetric titrations conducted with all nine pentapeptides (Appendix A) and at different analysis temperatures did not show any sign of binding. This was an unexpected finding, since CheB_Pec shares 86% sequence identity with the *E. coli* CheB that was shown to bind in its unphosphorylated form the free or receptor-born NWETF pentapeptide [15,16,17]. To verify whether the N-terminal His-tag at CheB_Pec may potentially prevent binding, the His-tag was enzymatically removed from CheB, but no binding was observed in isothermal titration calorimetry (ITC). Since the phosphorylation of *E. coli* CheB greatly enhanced its methylesterase activity [17], we hypothesized that phosphorylation may be a necessary prerequisite for pentapeptide binding to CheB_Pec. To verify this hypothesis, we generated purified CheB_Pec containing a beryllium fluoride adduct that mimics phosphorylation [45,52]. However, microcalorimetric titrations did not evidence binding.

### 2.5. Three-Dimensional Structure of a Pentapeptide-Independent CheB Methylesterase

The above results suggest that the single CheB in a strain that harbors 19 chemoreceptors with a pentapeptide is unable to bind any of these pentapeptides. One possible explanation may be that the protein is unfolded or present in an inactive conformation. To address this issue, we crystallized CheB_Pec and solved its three-dimensional structure at a resolution of 2.3 Å (Figure 6).

The enzyme is composed of an N-terminal REC domain and a C-terminal methylesterase domain that are connected by a linker of approximately 25 amino acids (Figure 6). The asymmetric unit contains five CheB_Pec chains that can be closely superimposed onto each other, resulting in Cα root mean square deviation (RMSD) values of 0.23 to 0.61 Å (Appendix A). These five chains can also be closely aligned with the structure of *S. enterica* sv. Typhimurium CheB [20,23], as evidenced by RMSD values between 1.21 to 1.47 Å (Appendix A), as well as onto the receiver and catalytic domains of *Thermotoga maritima* CheB (PDB ID: 3t8y and 3sft). In each CheB_Pec chain, there was a gap due to a lacking electron density that, depending on the chain, extended from amino acids 137 to 144–150. Lacking electron density is generally attributed to the corresponding protein segment being disordered. The gap was flanked by a segment with high B-factor values, indicative of significant protein flexibility in this region [54] (Appendix A). 

There was a very satisfactory overall structural alignment of the CheB structures from *S. enterica* sv. Typhimurium and *P. atrosepticum* SCRI1043 (Figure 7A). Major deviations in this alignment showed a small region in the receiver domain, as well as in both flanking regions of the gap (Appendix A). The segment in the *S. enterica* sv. Typhimurium structure corresponding to the gap in CheB_Pec was characterized by low mean B-factors, namely 20 ± 7 for chain A and 30 ± 9 for chain B, indicative of a well-ordered structure. 

### 2.6. The Region Corresponding to the Pentapeptide-Binding Site in E. coli CheB Is Disordered in P. atrosepticum CheB

Having provided evidence that CheB_Pec is a correctly folded protein that resembles closely the *S. enterica* sv. Typhimurium structure, the question as to why it does not bind pentapeptides remained. The answer to this question may be related to studies that have identified the pentapeptide-binding site at *E. coli* CheB. This binding site was found to comprise amino acids 130 to 140 (colored in red in Figure 7A,B) and is located on the C-terminal extension of the REC domain and N-terminal part of the linker [16]. The inspection of the sequence alignment of CheB from *E. coli* and *P. atrosepticum* SCRI1043 revealed a high degree of sequence divergence in the pentapeptide-binding area (Figure 7B). Importantly, a large part of the *E. coli* CheB pentapeptide-binding site overlaps with the gap observed in the CheB_Pec structure (shaded in grey in Figure 7B). We therefore hypothesize that the structural disorder of CheB_Pec in the region homologous to the pentapeptide-binding site in *E. coli* may be related to the failure to bind pentapeptides. In the case of CheR, distinct sequence features were identified for the pentapeptide-dependent and independent forms [8,13]. In contrast, the sequence alignment of pentapeptide-dependent and independent CheB did not reveal any obvious conserved sequence features.

## 3. Discussion

Many bacteria contain multiple paralogs of signaling proteins that form part of different chemosensory pathways [1]. A central question is whether or to what degree there is a specificity of interactions between the different homologs of signaling proteins and chemoreceptors. Furthermore, bacteria contain frequently a significant number of chemoreceptors, of which some possess C-terminal pentapeptides that are generally considered additional binding sites for CheR and CheB [9]. *P. aeruginosa* has four chemosensory pathways and a single chemoreceptor that contains a C-terminal pentapeptide. We showed previously that exclusively CheR_2_ but not any of the remaining three CheR homologs of *P. aeruginosa* binds to the McpB pentapeptide [13]. Here, we show that the same holds for the four CheB homologs of *P. aeruginosa*, since CheB_2_ was the only homolog that interacted with the McpB-derived pentapeptide. The data thus show that, exclusively, the CheR and CheB homologs encoded by the Che_2_ gene cluster (Figure 1) bind to McpB, the only receptor predicted to stimulate the Che_2_ pathway [33]. This pathway is essential for the full virulence of *P. aeruginosa* [39,56], but its precise function still needs to be determined [37]. McpB and Che_2_ pathway homologs are widespread in pathogenic and nonpathogenic γ-Proteobacteria [57], suggesting a function that is not exclusively associated with virulence. Interestingly, this pentapeptide is present in most McpB homologs, and it was concluded that this motif represents a fundamental feature of the McpB-like family [57]. We propose that a major reason for the pentapeptide conservation is its capacity to bind the pathway-specific CheB_2_ and CheR_2_ homologs [57], corresponding to a mechanism permitting pathway isolation. The pathway isolation of two-component systems has been extensively studied, particularly in the Laub laboratory [58], but the corresponding knowledge for chemosensory pathways is scarce. The data available on *P. aeruginosa* indicate that these four pathways are isolated [33], and the findings of our study may represent one of the corresponding mechanisms. Genome analyses of bacteria with pentapeptide-containing chemoreceptors showed that strains containing a single pentapeptide-containing chemoreceptor are the most abundant (approx. 2500 genomes) [9]. Future research will show whether the corresponding pentapeptides exert a similar function in these species.

The dissociation constant for the binding of CheB_2_ to the McpB pentapeptide (93 µM) is well below the corresponding value determined for the CheR_2_-pentapeptide interaction of 0.52 µM [13]. However, the values for the CheR and CheB binding to the *P. aeruginosa* pentapeptide GWETF are in the same range as those reported for the *E. coli* CheB and CheR binding to the NWETF pentapeptide, namely *K*_D_ values of 2 µM [10] and 130 to 160 µM for the CheR [17] and CheB, respectively. This may suggest that the much lower pentapeptide affinity of CheB as compared to CheR may be a more general feature. Studies have so far shown that there are pentapeptide-dependent and pentapeptide-independent CheR methyltransferases [8,13,14,19]. Sequence and structural features in the CheR β-subdomain, responsible for pentapeptide binding, were identified that account for the capacity or incapacity to bind pentapeptides [8,13]. We show here that, in the analogy to CheR, CheB methylesterases also form pentapeptide-dependent and independent subfamilies. The reason for the failure of CheB_Pec to bind pentapeptides may be related to the structural disorder in the segment homologous to the pentapeptide-binding site in *E. coli* CheB. Bioinformatic studies are required to establish the evolutionary history of CheB proteins in Enterobacteria to assess which subfamily evolved first. However, in contrast to CheR, sequence analyses of this region in pentapeptide-dependent and independent CheBs did not permit to identify a feature that can be associated with the capacity to bind pentapeptides.

Chemotaxis is particularly relevant for the virulence of plant pathogens [59]. This is also reflected at the genome level, since more than 90% of plant pathogens, compared to 50% for the bacterial average, possess chemosensory pathways [59]. In addition, the average number of chemoreceptors in plant pathogens, 33, is well superior to the bacterial average of 14 [59,60]. In contrast to the relevance of chemotaxis in plant pathogens, there is little information on the corresponding molecular mechanisms. *P. atrosepticum* is among the top 10 plant pathogens [42] and a suitable model to study chemosensory signaling in a plant pathogen. The chemoreceptor repertoire of the strain SCRI1043 shows a number of unusual features: (1) Remarkably, 67% of its chemoreceptors possess a 4HB LBD, which is well above the bacterial average of approx. 31% [61]. This chemoreceptor family is characterized by its versatility, as reflected in the broad range of ligands recognized (i.e., amino acids, boric acid, inorganic phosphate, aromatic acids, citrate, etc.); its capacity to recognize ligands with high [62,63] and low specificity [29,64]; and its ability to bind ligands directly [51] or via ligand-binding proteins [65]. (2) Apart from the three Are-like receptors that possess cytosolic LBDs (Figure 4), there are no cytosolic chemoreceptors, suggesting that, primarily, extracellular signals are sensed. (3) The abundance of chemoreceptors with a C-terminal pentapeptide. More than 50% of *P. atrosepticum* SCRI1043 chemoreceptors contain pentapeptides, a number that is well above the bacterial average of 10% [9], suggesting that pentapeptide function is important for signaling. However, CheB function does not require pentapeptide binding, and future studies are necessary to elucidate to what degree CheR function requires pentapeptide recognition.

## 4. Materials and Methods

### 4.1. Bacterial Strains and Growth Conditions

Bacterial strains used in this study are listed in Table 1. *P. atrosepticum* SCRI1043 and its derivative strains were routinely grown at 30 °C in Luria broth (5-g/L yeast extract, 10-g/L bacto tryptone, and 5-g/L NaCl) or minimal medium (0.41-mM MgSO_4_, 7.56-mM (NH_4_)_2_SO_4_, 40-mM K_2_HPO_4_, and 15-mM KH_2_PO_4_) supplemented with 0.2% (*w*/*v*) glucose as the carbon source. *E. coli* strains were grown at 37 °C in LB. *E. coli* DH5α was used as a host for gene cloning. Media for the propagation of *E. coli* β2163 were supplemented with 300-mM 2,6-diaminopimelic acid. When appropriate, antibiotics were used at the following final concentrations (in μg mL^−1^): kanamycin, 50, tetracycline, 10, streptomycin, 50, and ampicillin, 100. Sucrose was added to a final concentration of 10% (*w*/*v*) when required to select derivatives that underwent a second crossover event during marker exchange mutagenesis.

### 4.2. Generation of Protein Expression Plasmids

Plasmids used in this study are listed in Table 1. Genes encoding *P. aeruginosa* PAO1 CheB_1_ (PA1459), CheB_2_ (PA0173), CheB_3_ (PA3703), CheB_4_ (PA0414), and *P. atrosepticum* SCRI1043 CheB (ECA1693) were amplified by PCR using the oligonucleotides indicated in Appendix A and genomic DNA as the template. The latter PCR product was digested with NheI and SalI, whereas the remaining products were digested with NdeI and BamHI. The resulting DNA fragments were cloned into pET28b(+) linearized with the respective endonucleases. The generated plasmids were verified by DNA sequencing.

### 4.3. Site-Directed Mutagenesis

The Hemsley method [72] was used to generate the CheB_2_ D55E mutant. The pair of overlapping mutagenic primers pET28_CheB_2__ D55E_f and pET28_CheB_2__ D55E_r (Appendix A) were used to amplify the entire plasmid pET28b-CheB_2_ using the *Pfu* Turbo DNA polymerase (Agilent Technologies, Santa Clara, CA, USA). Following the elimination of template DNA by a digestion with DpnI, the resulting mixture was transformed into *E. coli* DH5α, and colonies were selected on LB agar plates supplemented with kanamycin. Plasmid inserts and flanking regions were sequenced.

### 4.4. Protein Overexpression and Purification

Plasmids for the overexpression of the wt and mutant CheB proteins of PAO1 were transformed into *E. coli* BL21 (DE3). Alternatively, pET28b-CheB_Pec was transformed into *E. coli* BL21-AI^TM^. The resulting strains were grown under continuous stirring (200 rpm) at 30 °C in 2-L Erlenmeyer flasks containing 500 mL of LB medium supplemented with 50-μg/mL kanamycin. At an OD_660nm_ of 0.6, protein expression was induced by the addition of 0.1-mM isopropyl β-d-1-thiogalactopyranoside. In addition, L-arabinose was added to *E. coli* BL21-AI^TM^ cultures for a final concentration of 0.2% (*w*/*v*). Growth was continued at 16 °C overnight prior to cell harvest by centrifugation at 10,000× *g* for 30 min. Cell pellets for the purification of CheB proteins of *P. aeruginosa* were resuspended in buffer A (20-mM Tris/HCl, 500-mM NaCl, 5% (*v*/*v*) glycerol, 10-mM imidazole, 0.1-mM EDTA, and 5-mM β-mercaptoethanol, pH 8.0), whereas pellets for the purification of CheB_Pec were resuspended in buffer B (20-mM Tris/HCl, 150-mM NaCl, 10-mM imidazole, 0.1-mM EDTA, 10% (*v*/*v*) glycerol, and 10-mM β-mercaptoethanol, pH 8.0). Subsequently, cells were broken by French press treatment at 62.5 lb/in^2^. After centrifugation at 20,000× *g* for 30 min, supernatants were loaded onto 5-mL HisTrap HP columns (Amersham Biosciences, Little Chalfont, UK) equilibrated with the corresponding buffer A or B and eluted with an imidazole gradient of 40–500 mM in the corresponding buffer. When necessary, the His-tag was removed by treatment with bovine thrombin (Sigma-Aldrich, Buchs, Switzerland) at 2 U/mL and 16 °C for 2 h. For crystallization, CheB_Pec was dialyzed into 5-mM Tris/HCl, 5-mM Pipes, 5-mM Mes, 2-mM dithiothreitol, 150-mM NaCl, and 10% (*v*/*v*) glycerol, pH 7.4 and purified by size-exclusion chromatography using a HiPrep^TM^ 26/60 Sephacryl^TM^ S200 HR gel filtration column (GE Healthcare, Chicago, IL, USA) at a flow rate of 1 mL/min. All proteins were purified at 4 °C. Purified proteins were dialyzed overnight into the corresponding analysis buffers for immediate analysis.

### 4.5. Isothermal Titration Calorimetry (ITC)

All experiments were conducted on a VP microcalorimeter (Microcal, Amherst, MA, USA) at 25 °C. For the analysis of the CheB homologs of *P. aeruginosa*, 15–40 μM of proteins (dialyzed into 5-mM Tris/HCl, 5-mM Pipes, and 5-mM Mes, pH 7.0) were placed into the sample cell and titrated with 1–7-mM solution of the GWEEF peptide (synthesized by Biomedal S.L., Seville, Spain). For the analysis of CheB_Pec, the protein was dialyzed into 5-mM Tris/HCl, 5-mM PIPES, 5-mM MES, 10% (*v*/*v*) glycerol, 2-mM dithiothreitol, 150-mM NaCl, and 0.1-mM EDTA, pH 7.4, adjusted to 15–50 μM and titrated with 4.8–14.4-μL aliquots of 1–5-mM peptide solutions (synthesized by GenScript^®^, Piscataway, NJ, USA). All ligand solutions were prepared in dialysis buffer immediately before use. The mean enthalpies measured from the injection of the peptide into the buffer were subtracted from the raw titration data prior to data analysis with the “one binding site model” of the MicroCal version of ORIGIN.

### 4.6. Chemoreceptor Sequence Analysis

Sequences were retrieved from the MIST 3.0 database [73], transmembrane regions identified using DAS [74], and ligand-binding domains annotated according to Pfam [75]. Pentapeptides at chemoreceptors were identified as reported in [8]. These peptides matched the xZxxZ motif (where Z represents either F, W, or Y) and were separated from the chemoreceptor signaling domain by a linker sequence of at least 10 amino acids.

### 4.7. Derivatization of CheB_Pec by Beryllium Fluoride

A modified version of the protocol described in [45] was employed. Briefly, 0.1-M BeSO_4_, 10-mM NaF, and 10-mM MgCl_2_ (all final concentrations) were added to CheB_Pec dialyzed into 5-mM Tris/HCl, 5-mM PIPES, 5-mM MES, 10% (*v*/*v*) glycerol, 2-mM dithiothreitol, 150-mM NaCl, and 0.1-mM EDTA, pH 7.4. The resulting mixture was incubated at 25 °C for 3 h.

### 4.8. Construction of Mutants Deficient in cheA and cheB

Chromosomal mutants of SCRI1043 were constructed by homologous recombination using a derivative plasmid of the suicide vector pKNG101. These plasmids were confirmed by DNA sequencing and carried deletion mutant alleles for the replacement of wild-type genes in the chromosome. In all cases, plasmids for mutagenesis were transferred to *P. atrosepticum* SRCI1043 by biparental conjugation using *E. coli* β2163. The plasmids for the construction of the deletion mutants were generated by amplifying the up- and downstream flanking regions of the gene to be mutated. The resulting PCR products were digested with the enzymes specified in Table 1 and ligated in a three-way ligation into pUC18Not, producing plasmids pUC18Not_ΔcheB and pUC18Not_ΔcheA. Subsequently, the kanamycin resistance cassette Km3 from the plasmid p34S-km3 was inserted into the BamHI of pUC18Not_ΔcheA, resulting in plasmid pUC18Not_ΔcheA-Km3. The Δ*cheB* and Δ*cheA*-km3 deletion constructs were then subcloned into the marker exchange vector pKNG101 using NotI. Mutant strains defective in *cheA* and *cheB* were generated using plasmids pKNG101_ΔcheA-km3 and pKNG101_ΔcheB, respectively.

### 4.9. Quantitative Capillarity Chemotaxis Assays

Overnight cultures of *P. atrosepticum* were grown at 30 °C in minimal medium. At an OD_660_ of 0.35–0.4, the cultures were washed twice with chemotaxis buffer (50-mM K_2_HPO_4_/KH_2_PO_4_, 20-μM EDTA, and 0.05% (*v*/*v*) glycerol, pH 7.0) and diluted to an OD_660_ of 0.1 in the same buffer. Subsequently, 230 μL of the resulting bacterial suspension were place into the wells of 96-well plates. One-microliter capillary tubes (P1424, Microcaps; Drummond Scientific, Broomall, PA, USA) were heat-sealed at one end and filled with either the chemotaxis buffer (negative control) or chemotaxis buffer containing casamino acids. The capillaries were immersed into the bacterial suspensions at their open ends. After 30 min at room temperature, the capillaries were removed from the bacterial suspensions, rinsed with sterile water, and the content expelled into 1 mL of minimal medium salts. Serial dilutions were plated onto minimal medium supplemented with 15-mM glucose as the carbon source. The number of colony-forming units was determined after incubation at 30 °C for 36 h. In all cases, data were corrected with the number of cells that swam into buffer-containing capillaries.

### 4.10. Crystallization and Resolution of the Three-Dimensional Structure of CheB_Pec

CheB_Pec was dialyzed into 5-mM Tris/HCl, 150-mM NaCl, 2-mM dithiothreitol, and 10% (*v*/*v*) glycerol, pH 7.4 and concentrated to 2.35 mg/mL using 3-kDa cut-off centricon concentrators (Merckmillipore (Kenilworth, NJ, USA). Crystallization conditions were screened using the capillary counter-diffusion technique, and a set of conditions prepared ad hoc that were reported in [76]. CheB_Pec was loaded into capillaries of 0.2-mm inner diameters, and crystals appeared in several conditions, namely C4 (1.25-M sodium citrate and 0.1-M Na/HEPES, pH 7.5); C5 (1.7-M ammonium sulphate, 3.5% (*w*/*v*) PEG 400, and 0.1-M Na/HEPES, pH 7.5); and C7 (2.0-M ammonium sulphate and 0.1-M Tris/HCl, pH 8.0 and 8.5). Crystals were extracted from the capillary and equilibrated in the mother solution supplemented with 15% (*v*/*v*) glycerol. Individual crystals were placed into LithoLoops (Molecular Dimensions, Portobello, Sheffield, UK), flash-frozen in liquid nitrogen and stored until data collection at the Xaloc beamline of the Spanish Synchrotron Radiation Source Alba. Several full datasets were obtained and automatically indexed, reduced, and scaled using the default data processing with EDNA [77] within the MXCuBE [78] data collection interface. The automatically determined space groups, I422 or F422, did not permit phasing using molecular replacement. Data were manually inspected, indexed, and merged with iMOSFLM [79] in space group I4 and scaled and reduced using Aimless [80] of the CCP4 program suite [81]. The structure was determined by molecular replacement in Phaser [82] using a homology model [83] based on the CheB structure of *S. enterica* sv. Typhimurium (PDB ID 1a2o) and after lowering the symmetry of the space group to *I4*. Five polypeptide chains were placed into the unit cell giving rise to a Matthews coefficient [84] of 2.85 and a water content of 57%. Refinement was done by phenix.refine of the PHENIX suite [85], including titration-libration-screw (TLS) parameterization [86]. Cycles of manual building and inspection were done in Coot [87]. The final refined model was verified with Procheck [88], Molprobity [89], and PDBe validation servers [90]. Appendix A summarizes the crystallographic data statistics and final model characteristics. The coordinates and the experimental structure factors for the CheB methylesterase from *P. atrosepticum* SCRI1043 were deposited at the Protein Data Bank with ID 6ymz.

## Figures and Tables

**Figure 1 ijms-21-08459-f001:**
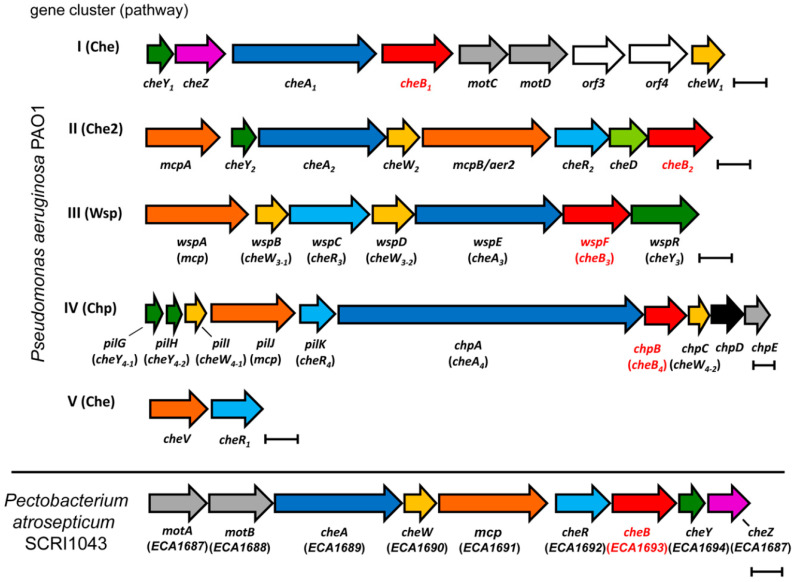
Gene clusters encoding chemosensory signaling proteins in *Pseudomonas aeruginosa* PAO1 and *Pectobacterium atrosepticum* SCRI1043. Genes of the same family are colored in the same color. The genes of the proteins studied in this article are shown in red. Bars, 0.5 kbp.

**Figure 2 ijms-21-08459-f002:**
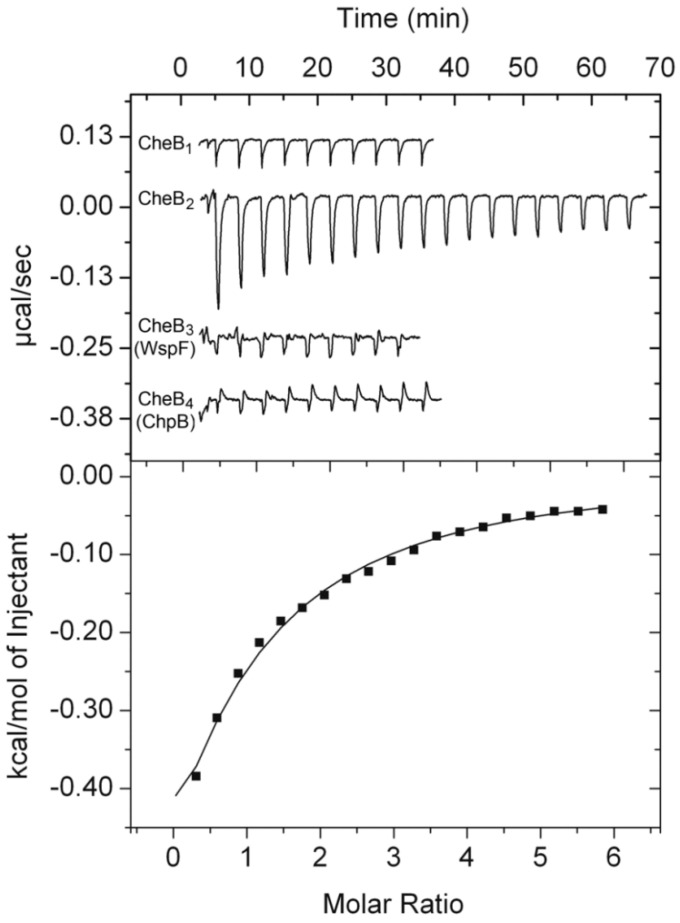
Specificity of the interaction between four CheB homologs of *P. aeruginosa* and the terminal pentapeptide GWEEF of the McpB chemoreceptor. Microcalorimetric titrations of the four CheB homologs (15 to 40 μM) with 14-μL aliquots (1 to 7 mM) of the GWEEF pentapeptide. Upper panel: raw titration data; lower panel: integrated, dilution heat-corrected and concentration-normalized peak areas of the titration data for CheB_2_. Data were fitted using the “one binding site model” of the MicroCal version of ORIGIN.

**Figure 3 ijms-21-08459-f003:**
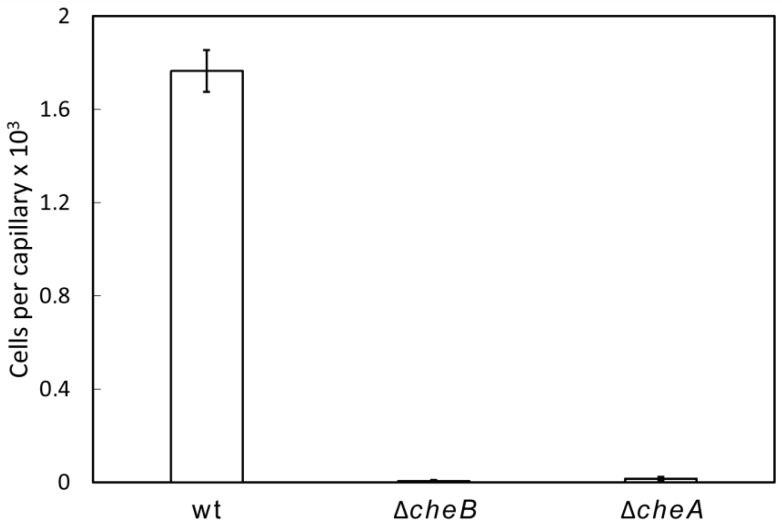
Quantitative capillary chemotaxis assays of wild-type and mutant strains of *P. atrosepticum* SCRI1043 towards 0.1% (*w*/*v*) casamino acids. Data were corrected with the bacteria that migrated into buffer-containing capillaries (225 ± 35). Data are means and standard deviations from three experiments conducted in triplicate. wt: wild-type.

**Figure 4 ijms-21-08459-f004:**
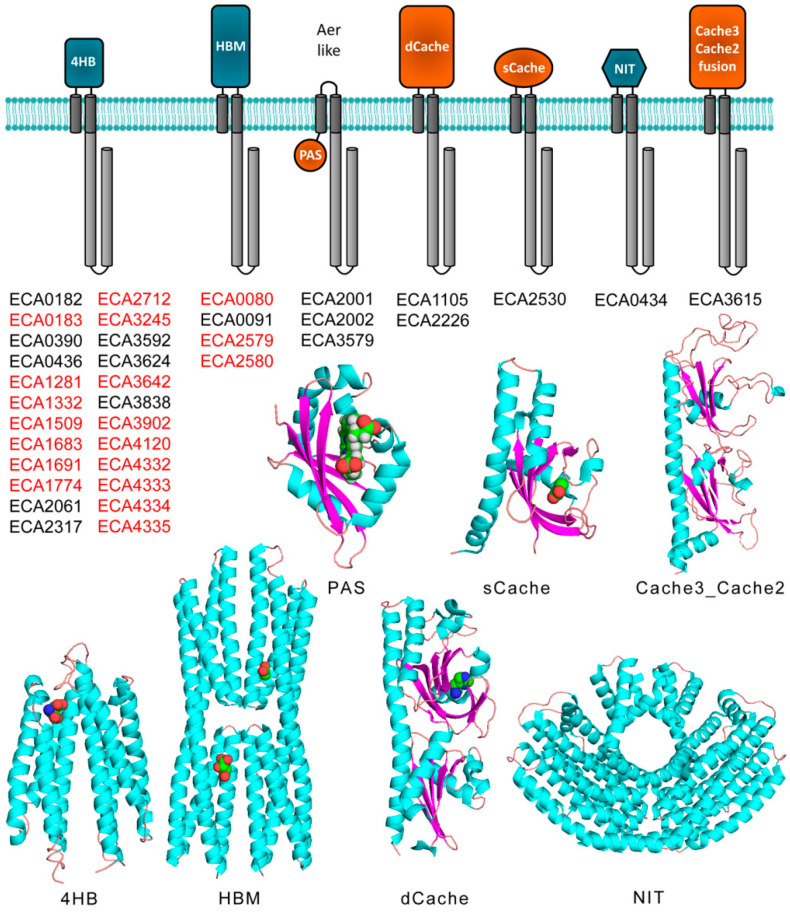
The chemoreceptor repertoire of *P. atrosepticum* SCRI1043. Ligand-binding domains with α/β folds or parallel helices are shown in orange or blue, respectively. Chemoreceptor names in red indicate receptors with C-terminal pentapeptides. Shown below are representative 3D structures of these domains, namely the structure of Tar-LBD (4HB) in a complex with aspartate (PDB ID: 1vlt), McpS-LBD (HBM) in a complex with malate and acetate (PDB ID: 2yfa), Aer2-LBD (PAS) in a complex with heme (PDB ID: 4hi4), TlpQ-LBD (dCache) in a complex with histamine (PDB ID: 6fu4), PscD-LBD (sCache) in a complex with propionate (PDB ID: 5g4z), NasR-LBD (NIT) (PDB ID: 4akk), and a homology model of the ECA3615-LBD (Cache3_Cache2 fusion) generated by SwissModel [48] using PDB ID 4avf as template.

**Figure 5 ijms-21-08459-f005:**
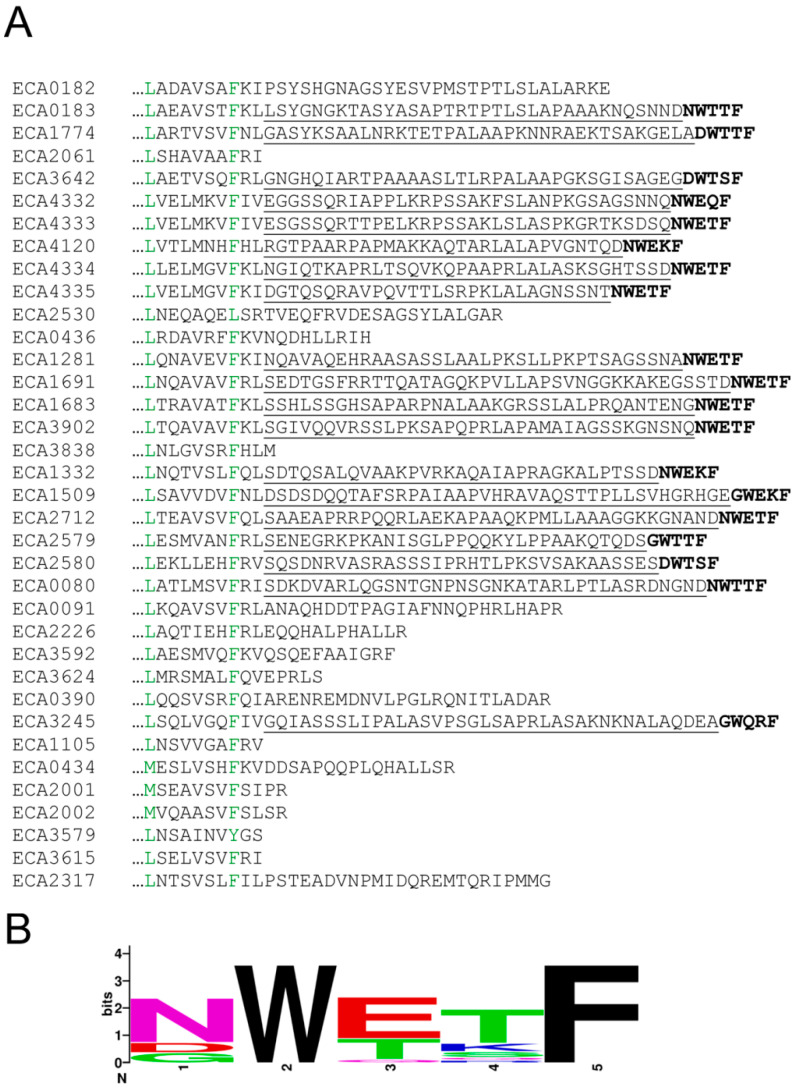
C-terminal pentapeptides at *P. atrosepticum* chemoreceptors. (**A**) C-terminal section of the sequence alignment of *P. atrosepticum* SCRI1043 chemoreceptors. Pentapeptides are in boldface, and the linker sequences are underlined. Residues in green are highly similar. (**B**) Sequence logo of the 19 pentapeptides. The figure was generated using Weblogo (https://weblogo.berkeley.edu/logo.cgi).

**Figure 6 ijms-21-08459-f006:**
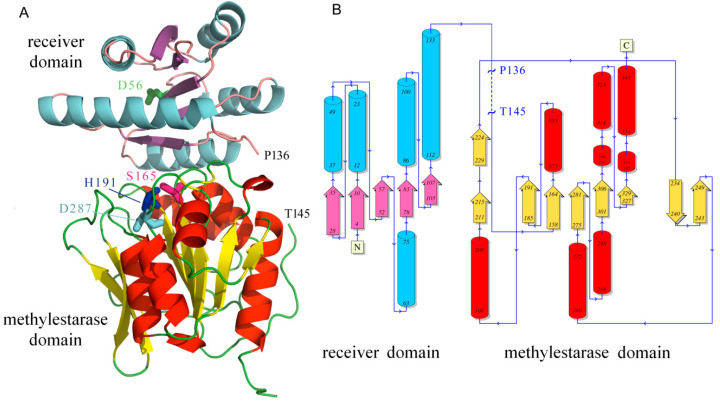
The three dimensional structure of *P. atrosepticum* CheB. (**A**) Ribbon diagram of the structure. The gap observed due to lacking electron density is indicated (P136 to T145). The phosphoryl group accepting aspartate (D56), as well as the residues that form the methylesterase catalytic triad (S165-H191-D287), are shown in stick mode. (**B**) Secondary structure elements: tubes: α-helix; arrows: β-strand. The figure was produced using PDBsum [53].

**Figure 7 ijms-21-08459-f007:**
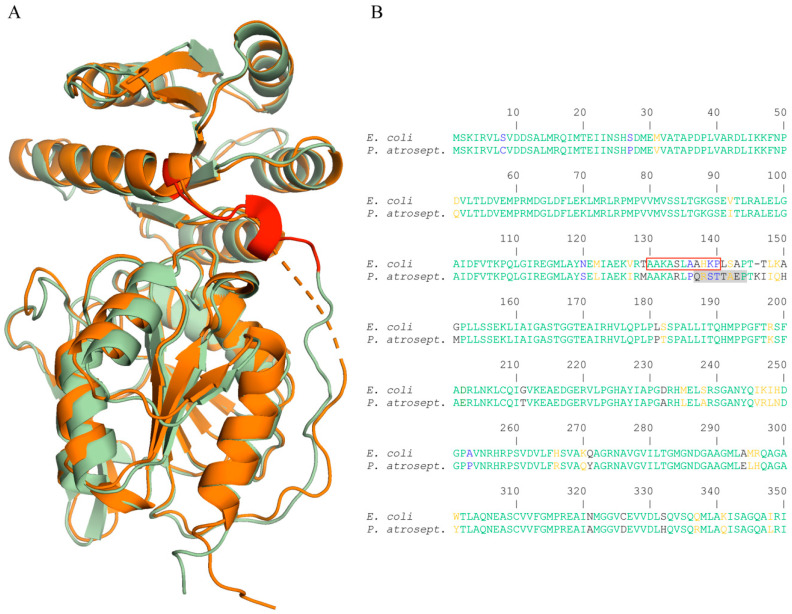
Structural and sequence features related to the capacity of CheB to recognize C-terminal pentapeptides. (**A**) Structural alignment of CheB from *P. atrosepticum* (orange) and *S. enterica* sv. Typhimurium (green, PDB ID: 1A2O). The amino acid segment identified as being the pentapeptide-binding site [16] is shown in red. (**B**) Sequence alignment of *Escherichia coli* K-12 and *P. atrosepticum* SCRI1043 CheB. The amino acids that form the pentapeptide-binding site in the *E. coli* enzyme are boxed in red. The gap in the CheB_Pec structure is shaded in grey. The alignment was done using the CLUSTALW algorithm of the NPS@ software [55]. The Gonnet protein weight matrix was used; gap opening and gap extension penalties were 10.0 and 0.1, respectively. Residues in green are identical, orange highly similar, blue weakly similar, and black dissimilar.

**Table 1 ijms-21-08459-t001:** Strains and plasmids used in this study.

Strains and Plasmids	Genotype or Relevant Characteristics ^a^	Reference
**Strains**		
*Escherichia coli* BL21(DE3)	F^−^ *ompT gal dcm lon hsdS_B_*(*r_B_*^−^*m_B_*^−^) λ(DE3 (*lacI lacUV5*-*T7p07 ind1 sam7 nin5*)) (*malB*^+^)_K-12_(λ^S^)	[66]
*E. coli* BL21-AI	F- *ompT hsdS*_B_ (r_B_^−^m_B_^−^) *gal dcm araB*::*T7RNAP-tetA*	Invitrogen
*E. coli* DH5α	F^−^ *endA1 glnV44 thi-1 recA1 relA1 gyrA96 deoR nupG purB20* φ80d*lacZ*ΔM15 Δ(*lacZYA-argF*)U169, hsdR17(*r_K_*^−^*m_K_*^+^), λ^−^	[67]
*E. coli* CC118λ*pir*	*araD* Δ(*ara*, *leu*) Δ*lacZ74 phoA20 galK thi-1 rspE* *rpoB argE recA1* λ*pir*	[68]
*E. coli* β2163	F- RP4-2-Tc::Mu Δ*dapA*::(*erm-pir*); Km^R^, Em^R^	[69]
*Pectobacterium atrosepticum* SCRI1043	Wild type strain	[32]
*P. atrosepticum* SCRI1043 Δ*cheB*	SCRI1043 in-frame deletion mutant of *cheB*	This study
*P. atrosepticum* SCRI1043 Δ*cheA*	SCRI1043 deletion mutant of *cheA*; Km^R^	This study
**Plasmids**		
pET28b(+)	Protein expression plasmid; Km^R^	Merckmillipore (Kenilworth, NJ, USA)
pET28b-CheB_1_	Km^R^; pET28b(+) derivative containing *P. aeruginosa cheB1* (PA1459)	This study
pET28b-CheB_2_	Km^R^; pET28b(+) derivative containing *P. aeruginosa cheB2* (PA0173)	This study
pET28b-CheB_2_ D55E	Km^R^; pET28b(+) derivative containing *P. aeruginosa cheB2* D55E mutant	
pET28b-CheB_3_	Km^R^; pET28b(+) derivative containing *P. aeruginosa cheB3* (PA3703)	This study
pET28b-CheB_4_	Km^R^; pET28b(+) derivative containing *P. aeruginosa cheB4* (PA0414)	This study
pET28b-CheB_Pec	Km^R^; pET28b(+) derivative containing *P. atrosepticum cheB* (ECA1693)	This study
pUC18Not	Ap^R^; identical to pUC18 but with two NotI sites flanking pUC18 polylinker	[68]
pUC18Not_ΔcheB	Ap^R^; 1.5-kb PCR product containing a 954 bp in frame deletion of *cheB* (ECA1693) of SCRI1043 inserted into the SphI/NdeI sites of pUC18Not	This study
pUC18Not_ΔcheA	Ap^R^; 1.5-kb PCR product containing a 1561 bp deletion of *cheA* (ECA1689) of SCRI1043 inserted into the EcoRI/HindIII sites of pUC18Not	This study
p34S-Km3	Km^R^, Ap^R^; *Km3* antibiotic cassette	[70]
pUC18Not_ΔcheA-km3	Ap^R^, Km^R^; 0.96-kb BamHI fragment containing *km3* cassette of p34S-Km3 was inserted into BamHI site of Δ*cheA* in pUC18Not_ΔcheA	This study
pKNG101	Sm^R^; *oriR6K mob sacBR*	[71]
pKNG101_ΔcheB	Sm^R^, Km^R^; 1.5 kb NotI fragment of pUC18Not_ΔcheB was cloned at the same site in pKNG101	This study
pKNG101_ΔcheA-km3	Sm^R^, Km^R^; 2.4-kb NotI fragment of pUC18Not_ΔcheA-Km3 was cloned at the same site in pKNG101	This study

^a^ Ap, ampicillin, Em, erythromycin, Km, kanamycin, Sm, streptomycin, and Tc, tetracycline.

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
