# Peer review of "Evidence for Pentapeptide-Dependent and Independent CheB Methylesterases"

_ijms, 2020, doi:10.3390/ijms21228459_

Round 1

Reviewer 1 Report

In this paper by Velando et al, the authors investigate the binding specificity of the chemotaxis-regulating CheB methyltransferase. CheB binds to chemotaxis receptors in both C-terminal pentapeptide-dependent and -independent manners. Previous work on CheR identified two classes of these enzymes which differ in their pentapeptide binding. In this paper, the authors look for pentapeptide specificity among the 4 different chemotaxis systems in P. aeruginosa PAO1 and found that CheB2 specifically binds to pentapeptides. They also examine the sole CheB in P. atrosepticum and find that it does not bind pentapeptides despite 19 receptors in that organism that have a pentapeptide domain. The authors solved the crystal structure of the P. atrosepticum CheB and identified an unstructured region adjacent to the predicted pentapeptide-binding domain, based on homology to E. coli CheB. They conclude that this disordered region gives rise to the inability to bind pentapeptides.

This paper addresses a very interesting question in determining how different chemotaxis systems isolate from one another, potentially through the role of pentapeptide binding. In general, I thought the data were quite clear and the experiments were logical. The only area of concern I have is regarding the conclusion that the disordered domain in P. atrosepticum CheB is the cause of its inability to bind pentapeptides.

That conclusion is made by comparing the sequence to E. coli CheB. However, just to do a quick check, I did an alignment of P. aeruginosa CheB2 and P. atrosepticum CheB and find that they are quite similar in their proposed binding domains. Further, using a prediction of B-factors shows that the PAO1 CheB2 is similarly disordered in this region, yet you have shown that it binds pentapeptide with reasonable affinity. Therefore, it seems that disorder in this region may not regulate pentapeptide binding.

To address this discrepancy, I would suggest several experiments:

  1. Swap the ordered domain from coli into the disordered domain of P. atrosepticum to see if pentapeptide binding is increased.
  2. Swap the disordered domain from PAO1 into atrosepticum to see if that increases binding.
  3. Try the glutamine point mutation rather than beryllium fluoride to mimic phosphorylation.

Author Response

In this paper by Velando et al, the authors investigate the binding specificity of the chemotaxis-regulating CheB methyltransferase. CheB binds to chemotaxis receptors in both C-terminal pentapeptide-dependent and -independent manners. Previous work on CheR identified two classes of these enzymes which differ in their pentapeptide binding. In this paper, the authors look for pentapeptide specificity among the 4 different chemotaxis systems in P. aeruginosa PAO1 and found that CheB2 specifically binds to pentapeptides. They also examine the sole CheB in P. atrosepticum and find that it does not bind pentapeptides despite 19 receptors in that organism that have a pentapeptide domain. The authors solved the crystal structure of the P. atrosepticum CheB and identified an unstructured region adjacent to the predicted pentapeptide-binding domain, based on homology to E. coli CheB. They conclude that this disordered region gives rise to the inability to bind pentapeptides.

This paper addresses a very interesting question in determining how different chemotaxis systems isolate from one another, potentially through the role of pentapeptide binding. In general, I thought the data were quite clear and the experiments were logical. The only area of concern I have is regarding the conclusion that the disordered domain in P. atrosepticum CheB is the cause of its inability to bind pentapeptides.

That conclusion is made by comparing the sequence to E. coli CheB. However, just to do a quick check, I did an alignment of P. aeruginosa CheB2 and P. atrosepticum CheB and find that they are quite similar in their proposed binding domains. Further, using a prediction of B-factors shows that the PAO1 CheB2 is similarly disordered in this region, yet you have shown that it binds pentapeptide with reasonable affinity. Therefore, it seems that disorder in this region may not regulate pentapeptide binding.

Response: Many thanks for these favorable comments. We agree with this referee in that the data presented may not permit to conclude. However, we feel that the finding that the region equivalent to the pentapeptide binding site in E. coli CheB is disordered in CheB_Pec permits to hypothesize that this may be related to the inability to CheB_Pec to bind pentapeptides. We have changed the manuscript and the “conclusion” of the initial version of the manuscript is now labelled as a “hypothesis”.  

To address this discrepancy, I would suggest several experiments:

  1. Swap the ordered domain from coli into the disordered domain of P. atrosepticum to see if pentapeptide binding is increased.
  2. Swap the disordered domain from PAO1 into atrosepticum to see if that increases binding.
  3. Try the glutamine point mutation rather than beryllium fluoride to mimic phosphorylation.

Response: Thank you very much for these interesting comments. This is something I would like to pursue in my laboratory and I am currently applying for money to do so. The editor has asked me to re-submit a revised version of this manuscript within 10 days, which does not permit to carry out any additional experimental work.

Reviewer 2 Report

In this study, Velando et al. describe their results on chemosensory pathways from Pseudomonas aeruginosa and Pectobacterium atrosepticum, and the crystal structure of the CheB ortholog from P. atrosepticum that gives structural insights into the absence of pentapeptide binding of this enzyme.

The aims of the study are clearly stated, methods are well described and data are well presented and discussed. I agree with the publication of the manuscript, with the following minor revisions:

  • L140: the previously measured affinity of the interaction between CheR2 and the peptide (0.52mM) should be specified.
  • L179 and L322: the author should clarify whether Aer-like receptors are cytosolic or not.
  • L210: references 54-56 should be substituted by references 52-53.
  • Fig6B: the software used to generate the secondary structure diagram should be specified.
  • L231: ‘unstructured’ should be substituted by ‘disordered’.
  • L442: ‘I422 of F422 ’ should be substituted by ‘I422 or F422’.

Author Response

In this study, Velando et al. describe their results on chemosensory pathways from Pseudomonas aeruginosa and Pectobacterium atrosepticum, and the crystal structure of the CheB ortholog from P. atrosepticum that gives structural insights into the absence of pentapeptide binding of this enzyme.

The aims of the study are clearly stated, methods are well described and data are well presented and discussed. I agree with the publication of the manuscript, with the following minor revisions:

Response: Thank you very much for your favorable comments.

L140: the previously measured affinity of the interaction between CheR2 and the peptide (0.52mM) should be specified.

Response: As suggested the dissociation constant of 0.52 microM is now provided.

L179 and L322: the author should clarify whether Aer-like receptors are cytosolic or not.

Response: On line 179 we have added “(Fig. 4)” after the statement “There are three receptors that possess the typical topology and domain arrangement of Aer receptors [49] that mediate aerotaxis.” In this Figure the three receptors are labelled as “Aer like”. As shown in this Figure receptors are membrane bound but possess cytosolic sensor domains.

The section on line 322 has been changed into “Apart from the three Are like receptors that possess cytosolic LBDs (Fig. 4), there are no cytosolic chemoreceptors suggesting that primarily extracellular signals are sensed.” This expresses more precisely the fact that P. atrosepticum has mainly receptors that sense signals in the extracytosolic space.

L210: references 54-56 should be substituted by references 52-53.

Response: Thanks. Done.

Fig6B: the software used to generate the secondary structure diagram should be specified.

Response: Thanks. The software and the corresponding reference are now indicated.

L231: ‘unstructured’ should be substituted by ‘disordered’.

Response: I fully agree that “disordered” is more precise. The same change has also been made in the heading of section 2.6.

L442: ‘I422 of F422 ’ should be substituted by ‘I422 or F422’.

Response: Thanks. Done.

Reviewer 3 Report

The Authors suggested that certain peptide sequences are critical to binding to chemosensory proteins. The hypothesis is sound but the proof lacks soundness.

  1. Peptide fragments may bind to almost any protein at sufficiently high concentrations. Due to experimental limitations, the binding affinity may or may not be determined.
  2. Figure 2 shows a clear discrimination between proposed binders. Only 1 protein binds the peptide which is a sign of a specificity. However, stoichiometry of binding shows that the process uses most likely multiple sites (6 at least), negating a hypothesis that a single site is responsible. Despite that, the Authors claim that the purported binding sequence is as deduced from the alignment and very hydrophobic. As expected, the Authors were not able to determine binding parameters due to technical problems (solubility).
  3. The Authors continue on their road to "prove" that the peptide binds to the protein by crystallizing the target and aligning structures to find the putative binding site. Since the sequence in question is random and have any conformation, cocrystallization with the peptide is out of question.

The work needs a substantial improvement before claims can be made. First, there should be a potential complex co-crystallized composed of 2 full proteins. It is likely that the peptide may be critical for some contacts/conformational changes that are important for complex formation. It may also be a non-specific binder using protein surface as the concentration surface before a more important interaction with other proteins takes place.

Language, presentation and graphics quality are very good. The manuscript needs a conceptual change and the supporting data to make the claim.

Author Response

The Authors suggested that certain peptide sequences are critical to binding to chemosensory proteins. The hypothesis is sound but the proof lacks soundness.

  1. Peptide fragments may bind to almost any protein at sufficiently high concentrations. Due to experimental limitations, the binding affinity may or may not be determined.

Response: We only partially agree with this referee. Peptide-protein interactions with KD values in the micromolar range are considered to be physiologically relevant. In analogy to the interaction of the NWETF pentapeptide with E. coli CheB (KD of 130 to 160 micromolar) that was shown to be of physiological relevance, we argue that the interaction between the GWETF pentapeptide and CheB2 (KD of 93 micromolar) is equally of physiological relevance. In four occasions (titrations of P. aeruginosa CheB1, CheB3 and CheB4 as well as titration with CheB_Pec with different pentapeptides) we did not observe any binding heats. These experiments involved a titration of the protein with a peptide solution (placed in the injector syringe) of 1 to 7 mM. In all four occasion, not even an onset of a binding interaction was noted since peak areas were small and uniform and comparable in its size to those resulting from a titration of dialysis buffer with the same concentration of pentapeptides. From these experiments we can conclude that no binding occurs with KD values in the micromolar range. We are unable to rule out that there is an interaction with an elevated KD such as several mM; however, there is a general consensus in the scientific community that interaction with such equilibrium constants are unlikely to be of mechanistic relevance.

  1. Figure 2 shows a clear discrimination between proposed binders. Only 1 protein binds the peptide which is a sign of a specificity. However, stoichiometry of binding shows that the process uses most likely multiple sites (6 at least), negating a hypothesis that a single site is responsible. Despite that, the Authors claim that the purported binding sequence is as deduced from the alignment and very hydrophobic. As expected, the Authors were not able to determine binding parameters due to technical problems (solubility).

Response: We beg to differ with this referee. The titration of CheB2 with the pentapeptide was conducted until a peptide-protein ratio in the sample cell of 6, which is a concentration close to the completion of CheB2 saturation with peptide. We imagine that the referee has taken this value of 6 to state that there are multiple, at least 6 different binding sites. However, the reasoning employed by this reviewer in incorrect. The visual inspection of hyperbolic ITC binding curves does not permit to draw any conclusion on the stoichiometry of interaction. The measurement of lower affinity binding events (KD value significantly above the macromolecule concentration in the sample cell) by ITC implies that the majority of ligand injected into the protein containing sample cell does not interact with the protein but remains unbound. This gives rise to hyperbolic shape curves, but also implies that ratios of ligand to sample concentrations, as shown on the lower X-axis of Fig. 2, are irrelevant for making any conclusion on the binding stoichiometry. This is in stark contrast to high affinity binding events (KD value significantly below the macromolecule concentration in the sample cell). In this binding mode the majority of ligand injected binds to the macromolecule giving rise to sigmoidal curves. The steep part of the sigmoidal curve represents thus rapid saturation of the sample cell ligand reflecting macromolecule depletion in the sample cell. As a consequence only from sigmoidal but not hyperbolic curves the binding stoichiometry can be estimated (not calculated), which corresponds to the ligand to macromolecule ratio at the point of inflection of the sigmoidal curve.

Two parameters permit getting insight into binding stoichiometry from hyperbolic binding curves. Firstly, data fitting gives rise to the n parameter, representing stoichiometry. However, the last author of this manuscript has been using ITC for more than 25 years and has published some 80 articles with ITC data. In his experience the n value determined by curve fitting is highly unrelieable since it corresponds to an extrapolation. For information, the n value determined in the experiment shown was 0.5. Apart from the above mentioned imprecision in determining the n values from hyperbolic curves, an underestimation of the binding stoichiometry is frequently due to inactive protein in the protein sample.

The other parameter that permits to gain insight into the binding stoichiometry is the quality of the curve fit. Data presented in Fig. 2 were fitted with a model for the binding of a single ligand to a single site at the macromolecule. The quality of curve fit and error statistics of data shown in Fig. 2 are entirely satisfactory (chi square of 56) and fully consistent with the binding of a single pentapeptide to CheB2.

What this reviewer refers to as technical problems are the above mentioned limitations imposed by the magnitude of ligand dilution heats. There was no problem with solubility.

  1. The Authors continue on their road to "prove" that the peptide binds to the protein by crystallizing the target and aligning structures to find the putative binding site. Since the sequence in question is random and have any conformation, cocrystallization with the peptide is out of question.

Response: We beg to differ in this issue. The reason for crystallizing and solving the structure of CheB_Pec was not to prove that the peptide binds to the protein. The motivation to crystallize CheB_Pec has been clearly stated “The above results suggest that the single CheB in a strain that harbours 19 chemoreceptors with a pentapeptide is unable to bind any of these pentapeptides. One possible explanation may be that the protein is unfolded or present in an inactive conformation. To address this issue, we crystallized CheB_Pec and solved its three dimensional structure at a resolution of 2.3 Å (Figure 6).” The failure to observe an expected protein activity may always be caused by protein un-or misfolding. In our opinion, the most adequate means to refute such a possibility is to solve the 3D structure of the protein. We were able to show that the structure of Che_Pect is highly similar to the CheB from S. typhiumurium; a finding that excludes the possibility that the lack of pentapeptide binding is due to protein un-or misfolding. Hopefully, other authors were equally careful in investigating the failure of expected ligand binding prior to drawing conclusions in the scientific literature.

The work needs a substantial improvement before claims can be made. First, there should be a potential complex co-crystallized composed of 2 full proteins. It is likely that the peptide may be critical for some contacts/conformational changes that are important for complex formation. It may also be a non-specific binder using protein surface as the concentration surface before a more important interaction with other proteins takes place.

Response: We do not understand what the referee means by “potential complex co-crystallized composed of 2 full proteins”. Which are the 2 full proteins? We are unable to understand the sentence “It may also be a non-specific binder using protein surface as the concentration surface before a more important interaction with other proteins takes place.”

Language, presentation and graphics quality are very good. The manuscript needs a conceptual change and the supporting data to make the claim.

Reviewer 4 Report

In the manuscript, the authors reported a new molecular basis for chemosensory pathway insulation. This is a good research article and should be of interest to the readership of the International Journal of Molecular Sciences. In my opinion, it just needs a minor revision.

Figure 2. Does it look like the authors only presented data from one experience? Could you try to present more experiences to get an error bar?

The authors generated a CheB2 D55E mutant. But I cannot see the authors presented any data about this mutant in the manuscript.

As original research, the authors put too many references.

Author Response

In the manuscript, the authors reported a new molecular basis for chemosensory pathway insulation. This is a good research article and should be of interest to the readership of the International Journal of Molecular Sciences. In my opinion, it just needs a minor revision.

Response: Thank you very much for your favorable comments.

Figure 2. Does it look like the authors only presented data from one experience? Could you try to present more experiences to get an error bar?

Response: We see the point of this referee. However, CheB2 is at the biochemical level not an easy protein. We have repeated the titration of CheB2 with the pentapeptide several times. However, although there was clear sign of binding, the titration data were often rather noisy and consequently the thermodynamic parameters associated with a significant error. We have thus decided against the calculation of means from multiple experiments and present the data obtained from the cleanest titration experiment, which is that shown in Fig. 2. We feel that the parameters shown are more precise than potential mean values.

The authors generated a CheB2 D55E mutant. But I cannot see the authors presented any data about this mutant in the manuscript.

Response: We mention these data at the end of section 2.1. where we state “Previous studies showed that the replacement of the phosphorylgroup accepting aspartate with glutamate in receiver domains mimics protein phosphorylation [46]. We have generated the CheB2 D55E mutant protein that was titrated with the GWEEF peptide resulting in a KD of 56 ± 14 µM (Figure S2), representing a modest increase in affinity as compared to the native protein.” The corresponding titration data are shown in Figure S2 of the Supplementary Material.

As original research, the authors put too many references.

Response: We agree with this referee. The number of references was reduced from 105 to 90. A significant number of references are necessary for the Materials and methods section and references to plasmids and strains.

Round 2

Reviewer 1 Report

I don't feel that my concerns about the significance of the disordered domain were appropriately considered. Given that the PAO1 CheB binds pentapeptide and seemingly has the same disordered domain as Pec CheB, cherry picking a comparison to E. coli doesn't make a compelling argument.

Reviewer 3 Report

The manuscript has been improved by inclusion of references, alternative explanation of discrepancies and other small changes. The small corrections are related to misuse, or lack, of hyphenation and can be corrected by a native English speaker.